# Experiences of women living with Polycystic Ovary Syndrome: A pilot case-control, single-cycle, daily Menstrual Cycle Diary study during the SARS-CoV-2 pandemic

Kaitlin Nelson[1,2,3], Sonia Shirin[1,2], Dharani Kalidasan[1], Jerilynn C. Prior[1,2,4,5]*

1 Centre for Menstrual Cycle and Ovulation Research, Division of Endocrinology and Metabolism, Department of Medicine, University of British Columbia, Vancouver, Canada, 2 BC Women's Health Research Institute, Vancouver, Canada, 3 Experimental Medicine, MSc Programme, University of British Columbia, Vancouver, Canada, 4 Division of Endocrinology and Metabolism, Department of Medicine, University of British Columbia, Vancouver, Canada, 5 School of Population and Public Health, University of British Columbia, Vancouver, Canada

* jerilynn.prior@ubc.ca

**Data Availability Statement:** Data cannot be shared publicly because we do not have ethical or

## Abstract

Polycystic Ovary Syndrome (PCOS) affects many people and is often distressing. Much medical literature about diagnosis and treatment exists, but little is known about PCOS menstrual cycle-related experiences except that cycles tend to be far-apart and unpredictable. Our purpose was to examine the menstrual cycle and daily life experiences in those with PCOS having approximately month-apart cycles compared with age and BMI-matched cohort controls using data from the Menstruation & Ovulation Study 2 (MOS2) during the first 1.5 years of SARS-CoV-2 pandemic. We hypothesized that those with PCOS would experience lower self-worth and more negative moods. This is a single-cycle prospective case-control study in community-dwelling women ages 19–35 years. Eight reported physician-diagnosed PCOS and were matched (1:3 ratio) with controls by age (within .6 years) and BMI (within .19 BMI units). Experiences were recorded daily (Menstrual Cycle Diary©, Diary). All kept daily morning temperatures to assess luteal phase lengths by the validated Quantitative Basal Temperature© analysis method. From 112 in MOS2, 32 women were compared: eight with PCOS versus 24 controls. Demographic, socioeconomic, comorbidities and lifestyle variables were not different between the two groups. Cycle lengths were similar in PCOS and controls (one PCOS and control each had oligomenorrhea; most lengths were 21–35 days, $P = .593$). Unexpectedly, luteal phase lengths were also similar between PCOS and controls ($P = .167$); anovulation occurred in 5 with PCOS, and in 9 controls. There were no significant Diary differences between the two groups except for greater "outside stress" in the PCOS group ($P = .020$). In contrast to our hypotheses, there were no significant differences in feelings of self-worth, anxiety nor depression. The SARS-CoV-2 pandemic was a stressful time for women. MOS2 captured granular menstrual cycles, ovulation and daily experiences in women with PCOS compared with age- and BMI-matched controls. These pilot data in women with milder PCOS are the first of more research required to understand the daily experiences in those living with PCOS.

legal permission from participants to make their individual data available. Data are available from Jerilynn C. Prior (jerilynn.prior@ubc.ca) or Nahid Shirazian - Institutional Data Access Contact (contact via (nahid.shirazian@ubc.ca) for researchers who meet the criteria for access to confidential data.

**Funding:** The author(s) received no specific funding for this work.

**Competing interests:** The authors have declared that no competing interests exist.

## Introduction

Polycystic Ovary Syndrome (PCOS) with androgen excess affects 1:10 in population-based cohorts of menstruating individuals [1]. PCOS, as appropriately diagnosed, is a multifaceted syndrome that affects not only women's and people with ovaries of reproductive age's (PORA) reproductive and metabolic health but also their mental health, quality of life and even eye anatomy and function [2]. However, we know little of the daily experiences of individuals with PCOS, especially those whose cycles have become more predictable and month-apart [3]. The Androgen Excess and Polycystic Ovary Syndrome Society utilizes the following criteria for diagnosis of Polycystic Ovary Syndrome (PCOS): 1) Hyperandrogenism: Presence of excessive male hormones (androgens), characterized by hirsutism (excessive hair growth) and/or hyper-androgenemia (high levels of androgens in the blood), 2) Ovarian dysfunction: Including irregular or absent menstrual periods (and/or oligo-anovulation) and 3) the presence of multiple ovarian cysts (polycystic ovaries) detected through imaging, with exclusion of other androgen excess or related disorders [4].

The medical-gynecological perspective dominates the literature surrounding PCOS. This extensive body of knowledge has worked to determine the etiology, genetic components, and management of PCOS. However, as a condition that alters women's lives, there are still many stories and experiences of PCOS yet to be explored [5]. It is important to note also, that the metabolic and clinical characteristics of individuals with PCOS may vary across geographical regions [6]. The call for greater diversity in populations is echoed in many research papers, emphasizing the need for culturally specific understanding and management recommendations for women living with PCOS [7, 8].

In the past few years, studies have been reporting the lived and living experiences of those with PCOS. Many studies have highlighted this syndrome's distressing effects on multiple aspects of life [7–10]. A review showed that Health-Related Quality of Life (HRQoL) scores were consistently reduced in those living with PCOS [11]. In an international survey in 1385 women (primarily from Europe and North America) the top concerns of women living with PCOS were 1) Difficulty losing weight, 2) Irregular menstrual cycles, 3) Infertility, and 4) Androgen excess (such as hirsutism) [12]. Those with PCOS also reported feeling that health-care providers did not understand PCOS and thus were limited in their abilities to provide appropriate education and treatment [12].

The negative impact of PCOS on the mental health of affected individuals is a significant concern. Those living with PCOS are more likely than other women of similar age and BMI to experience mental health issues [11, 13–16]. In 2018, the AE-PCOS Society issued a compelling call to action that emphasized the need to prioritize research on mental health among individuals with PCOS [11]. Such investigation has revealed that those with PCOS have a greater likelihood of experiencing moderate to severe symptoms of depression and anxiety [11]. Evidence suggested that high levels of stress were a critical determinant of both negative mood and anxiety problems in those with PCOS. This is potentially because these individuals may demonstrate significantly greater physiological reactions to stress compared to their healthy counterparts [17, 18], as well as higher rates of hospitalization due to stress and self-harming behaviour [19].

Despite extensive and increasingly important studies describing the concerns of women with PCOS [5], we know little of their daily life and menstrual cycle experiences compared with similar control women. Current studies of negative mood and stress in PCOS fail to include the typical menstrual cycle-related and daily life experiences of individuals living with PCOS. Diagnostic criteria for PCOS include longer menstrual cycles that are sometimes many months apart [11]. However, a comprehensive understanding of specific menstrual cycle-

related experiences as well as everyday PCOS experiences is essential for the development of effective PCOS-related health information resources. Although Shirin *et al's* (2021) recent study provided an in-depth Menstrual Cycle Diary© record of a single woman with PCOS's daily 6-cycles of experiences, it must be noted that this was a therapy study [20]. The only portion not likely influenced by Cyclic Progesterone Therapy was the first follicular phase [20]. Also, the data may not be representative of the typical experiences of women with PCOS because that woman began to report night sweats and hot flushes soon after that record was completed, thus was likely in the earliest phase of perimenopause [20].

Gaining a deeper understanding of the mood disorders, daily life experiences, and menstrual cycle-related experiences in those with PCOS, even though their condition may be less symptomatic than those with oligo-amenorrhea may have far-reaching positive implications. For example, greater understanding could lead to the identification of novel treatments to improve the quality of life for individuals with this condition. Additionally, it may promote advocacy for the needs of people with PCOS, thereby increasing resources and support for PCOS which involves a large portion of the premenopausal population. This knowledge can also help practitioners provide more effective and individualized care for those with PCOS. By recognizing the unique challenges faced by these individuals, healthcare professionals can tailor their treatments to better meet the needs of their patients. Overall, expanding our knowledge of the experiences of those with PCOS can lead to improved care, better outcomes, and a more comprehensive understanding of this complex condition.

The challenges of coping with PCOS as a chronic illness can lead to mental health issues that are exacerbated by external stressors such as limited access to healthcare services and social support, as Atkinson et al.'s 2021 study reported in women with PCOS during the COVID-19 pandemic [7]. Stress coping mechanisms are known to play a vital role in mediating the relationship between stress and mental disorders [21]. Coping strategies refer to an individual's cognitive and behavioral efforts to change, adjust, and interpret a stressful situation to reduce suffering. Active coping, which involves concentrating on the problem and striving to resolve it, is associated with protecting psychosocial well-being. Passive coping, by contrast, which involves concentrating on emotions and using strategies to reduce adverse emotions [22], is considered a maladaptive strategy that can worsen health-related quality of life and lead to anxiety and depression [18, 23].

Studies on the coping strategies of women with PCOS have been relatively limited until recently. Some studies indicate that women with PCOS use maladaptive coping processes [18, 23], although others suggest the opposite [24, 25]. Ego-resiliency is a personality trait that provides individuals with the necessary emotional, motivational, and cognitive resources to control their behavior and adapt to changing circumstances [21, 26]. It reduces the tendency to experience anxiety and depression even when a situation is perceived as stressful [27].

To better understand these dynamics, we looked for studies on the responses of women with PCOS to stressful situations. Dybciak's 2022 study found, in the 230 women with PCOS in a clinical cohort, that they had higher levels of anxiety and depression, poorer ego-resiliency, and used passive-coping strategies significantly more frequently than did the 199 healthy women chosen as convenience controls [28]. Socio-demographic variables such as living in a rural area, having a lower level of education and being childless were associated with increased anxiety levels [28]. In women with PCOS, depression was associated with being over 30 years old, living in a rural area, having lower levels of education, being childless, and being obese [28]. Low ego-resiliency and the use of passive coping strategies were also identified as predictors of high levels of anxiety and depression [28]. We know of no other similar analyses in women living with PCOS.

Our objective in this pilot study related to feasibility: 1) To determine, in a case-control study, if a woman with predictably cycling PCOS could effectively be matched with three age- and BMI-matched control women within a larger community-dwelling cohort; 2) To describe the menstrual cycle and daily experiences of women with PCOS having month-apart cycles compared with matched controls with recording daily in the standardized Menstrual Cycle Diary© [29].

## Materials and methods

### Cohort origins

In 2006, the Centre for Menstrual Cycle and Ovulation Research (CeMCOR) donated surplus menstrual cycle phase-specific urines samples from a large investigation [30] to Health Canada so they could measure environmental contaminant levels (Menstruation and Ovulation Study, MOS). Over a decade later, CeMCOR was funded by Health Canada to collect follow-up samples (http://hdl.handle.net/2429/76793). When designing the Menstruation and Ovulation Study 2 (MOS2), we pre-planned to assess menstrual cycle experiences and cycle and luteal phase lengths. The MOS2 study began recruitment in mid-February 2020. As it happened, few were enrolled before the first local SARS-CoV-2 lockdown began in March.

### Protocol

A secondary objective of the community-dwelling MOS2 cohort study protocol (http://hdl.handle.net/2429/76793), was to compare those self-reporting physician-diagnosed PCOS (cases) with age- and BMI- matched controls. Our intent was, for the first time, to compare respective menstrual cycle- and daily life-related experiences in these two groups. All MOS2 participants experienced changes and life disruptions of the COVID-19 pandemic. This study was approved by the Clinical Research Ethics Board of the University of British Columbia (*H19-02983*). Written, signed, and informed consent was obtained from all MOS2 study participants.

### Design

This study is a pilot prospective case-control study in community dwelling volunteers that was conducted across a single menstrual cycle, using data from 112 community-dwelling women/ PORA with all participants being between the ages of 19 and 35 years. Participants' eligibility for MOS2 required that they have approximately month-apart menstrual cycles and not be taking hormonal contraception. Thus, participants were excluded if they knew they had oligo-menorrhea, were using combined hormonal contraceptives (CHC), a hormone-releasing IUD or other ovarian hormonal therapies. The data were collected between February 15, 2020, and September 15, 2021. The data for this analysis were retrieved on August 18, 2022. The authors, except for one who was originally involved in data-collection, did not have access to information that could identify individual participants. Eight participants self-reported physician-diagnosed PCOS, met all other MOS2 eligibility criteria and were enrolled. We randomly matched each of these individuals living with normally cycling PCOS with three other cohort participants in a 1:3 ratio of cases to controls.

### Data collection tools

All participants completed a comprehensive baseline questionnaire administered by interviewers using the instrument from the Canadian Multicentre Osteoporosis Study [31], which provided social, educational, ethnicity, medical, comorbidity, reproductive, and lifestyle variables.

Comorbid conditions compared between groups included osteoporosis, osteoarthritis, endometriosis, infertility, heart diseases, and musculoskeletal diseases.

The Menstrual Cycle Diary© [29] (Diary), women's experiences-collection tool, could be compared with 1-year Diary normative data with 53 prospective normally ovulatory cohort participants with the most complete data [32] archived from the original investigation [33]. Participants were instructed to, each evening, record in their Diary the day's cycle-related and other everyday life experiences. This included ordinal scoring (0–4) for flow, cramps, breast tenderness, cervical mucus, sleep problems, and negative moods (frustration, depression and anxiety). Changes from expected in the bottom half of the Diary were scored on a 1–5 scale around the score of 3 which indicated their "usual" experience and included appetite, breast size, interest in sex, self-worth, energy and outside stressors.

Ovulation was confirmed using the twice-validated, non-invasive Quantitative Basal Temperature© (QBT©) method [34, 35]. Using QBT, normal luteal phase lengths average ~12 days but are at least 10 days long. Short luteal phases are documented when the QBT shows fewer than 10 days of elevated basal temperature. Anovulation is diagnosed when there is no significant temperature rise in the latter part of the cycle or a rise is present but for fewer than four days.

To ensure standardization, the researchers provided YouTube videos (https://www.cemcor.ca/resources/daily-menstrual-cycle-diary) to teach the Diary recording methods. In addition, all digital thermometers were supplied from the same batch and validated to be consistently accurate to 0.1-degree Celsius, thus minimizing measurement errors.

## Physical measurements

Height, weight, (providing BMI, calculated as $kg/M^2$), and waist circumference were collected by trained researchers using standard methods. This was true for the majority of participants. Because of the SARS-CoV-2 pandemic, 30 participants collected these data themselves using a provided tape measure and following detailed research instructions.

## Statistical analyses

**Parametric and non-parametric statistical analyses.** All data was assessed for normality of distribution. The physical measurements were analyzed with parametric methods given their normal distributions. However, the ordinal nature of the Diary data and the small numbers in this case-control study, led us to primarily use non-parametric statistical methods. Diary data were described, and hypotheses were tested using the non-parametric Mann-Whitney U, Chi-Square and Fisher's Exact tests. For Diary data relating to menstrual flow and cramps we reported both the number of days of flow and its mean daily intensity. We evaluated the controls and the cases by comparing their documented menstrual cycle and ovulatory characteristics. Ovulation could only be assessed in seven of the eight participants with PCOS —this occurred for technical reasons in a total of four of the 112 women participating in MOS2. We directly compared cases and control for cycle lengths and for ovulatory characteristics based on luteal phase lengths. For statistical analysis reasons only, in examining the mean luteal phase lengths in all cycles, we gave anovulatory cycles a "luteal length" of 0.1 days.

**Random selection of controls to match PCOS cases.** Given the total cohort of 112 and eight PCOS cases, we randomly first selected three controls per case based as closely as possible by years of age. We then checked to see if these controls were close in actual BMI value. We chose the best options for matching each of age (first) and BMI (second) variables for all cases compared with controls. Matching was close between cases and selected controls with half a year in age (0.6) mean difference (95% mean difference Cl -2.0; 3.2). In secondarily matching

for BMI (using it as a continuous variable) they were matched so that the mean BMI difference was 0.19 kg/M$^2$ (95% CI of that mean -4.6; 5.0).

All data were analyzed using SPSS Version 28.

## Results

The 32 participants (eight with normally cycling and milder PCOS and 24 controls) in this case-control pilot study averaged in their early 30s and were of similar ages (all in the 26 to 35-year range). All participants self-selected gender as "woman" (she/her). As shown in **Table 1** in categorical data, not only were the two case-control groups no different in age and

**Table 1. Demographic, anthropomorphic, medical, reproductive and lifestyle categorical characteristics of the cases with Polycystic Ovary Syndrome (PCOS) compared (1:3) with age-, body mass index-matched controls in the Menstruation Ovulation Study 2 (MOS2) community-dwelling cohort during the SARS-CoV-2 pandemic.**

| | PCOS (N = 8) | Control (N = 24) | Significance |
|---|---|---|---|
| | N (%) | N (%) | P^ |
| **Socio-Demographics and Lifestyle** | | | |
| **Body Mass Index** (kg/m$^2$) | | | .85 |
| Underweight ($<$ 18.5) | 1 (12.5) | 3 (12.5) | |
| Normal (18.5–24.9) | 2 (25.0) | 6 (25.0) | |
| Overweight (25.0–29.9) | 2 (25.0) | 10 (41.7) | |
| Obese ($\geq$ 30) | 3 (37.5) | 5 (20.8) | |
| **Ethnicity** | | | .741 |
| White | 5 (62.5) | 13 (54.2) | |
| East Asian (e.g. Chinese, Korean) | 2 (25.0) | 4 (16.7) | |
| Other or mixed | 1 (12.5) | 7 (29.2) | |
| **Occupation** | | | .482 |
| Fulltime | 3 (37.5) | 14 (58.3) | |
| Part-time | 4 (50.0) | 6 (25.0) | |
| Student | 1 (12.5) | 4 (16.7) | |
| **Living Situation** | | | .625 |
| Alone | 2 (25.0) | 4 (16.7) | |
| With $\geq$1 another adult (or children) | 6 (75.0) | 20 (83.3) | |
| **Current Cigarette Use** (% Yes) | 0 | 0 | |
| **Alcohol Use** (past 12 months) (% Yes) | 6 (75.0) | 15 (62.5) | .681 |
| **Regular Exercise Programme** (% Yes) | 3 (37.5) | 15 (62.5) | .252 |
| **Reproductive** | | | |
| **Nulliparous** (%Yes) | 6 (75) | 22 (91.7) | .254 |
| **Menstrual Cycle Length** | | | 1.000 |
| 21–35 days | 7 (87.5) | 22 (91.7) | |
| $>$ 35 days | 1 (12.5) | 2 (8.3) | |
| **Ovulation Category*** | | | .341 |
| Normal Ovulation | 1 (14.3) | 8 (33.3) | |
| Short Luteal Phase | 1 (14.3) | 7 (29.2) | |
| Anovulation | 5 (71.4) | 9 (37.5) | |
| **Comorbidities** | | | |
| **Musculoskeletal diseases** (% Yes) | 1 (12.5) | 1 (4.2) | .444 |

^Chi-Square test.

*Normally ovulatory cycles have a luteal phase length by Quantitative Basal Temperature of $\geq$10 days; Short luteal phase cycles have a luteal length of 4–9 days; Anovulatory cycles have no luteal phase.

BMI (on which they were matched), but all other demographic, anthropomorphic, socioeconomic, reproductive and comorbidity categorical variables also did not differ. Notably, the majority of participants, both cases (75%) and controls (83.3%), resided with at least one other adult or child/children. One individual from each case and control groups identified as having a musculoskeletal comorbidity, although differences were not significant; none identified any other comorbidity.

Table 2 shows the continuous data for age, socio-demographic, lifestyle, reproductive, and comorbidity characteristics between the eight women with PCOS and 24 control participants. No significant differences were found between the groups in age, height, weight, BMI, waist circumference, menstrual cycle or luteal phase lengths. The majority of participants in both groups had cycle lengths within the normal range of 21–35 days. However, the presence of subclinical ovulatory disturbances, including short luteal phases anovulation, were high in both cases (85.7%) and controls (66.7%).

Fig 1 shows that individuals living with PCOS in the MOS2 community cohort had similar cycle lengths as controls (likely because needed to have approximately month-apart cycles to be eligible). Cycle lengths in women with PCOS ranged from 21 to 111 days; in the controls cycle lengths were from 24 to 38 days. Although, the groups initially appeared to differ, the majority of both groups had normal-length cycles (21–35 days) and analysis showed a non-significant $P$ value of .593.

Women living with about month-apart cycling PCOS also had similar ovulatory characteristics as controls as indicated by luteal phase lengths (Fig 2). One participant with PCOS (of the seven with available ovulation data) and eight of the controls had normally ovulatory cycles. One participant with PCOS and seven controls had short luteal phases. Five participants with PCOS and nine controls recorded anovulatory cycles. These results showed that, in women with normally cycling PCOS, luteal phase lengths also did not significantly differ between cases and controls ($P$ = .167).

**Table 2. Age, anthropomorphic, exercise and reproductive continuous characteristics of the participants with Polycystic Ovary Syndrome (PCOS) compared (1:3) with age-, body mass index-matched controls all of whom had approximately month-apart cycles in the Menstruation Ovulation Study 2 (MOS2) Community dwelling cohort during the SARS-CoV-2 pandemic.**

| | PCOS (N = 8) | | Control (N = 24) | | Significance |
|---|---|---|---|---|---|
| | Mean ± SD | 95% CI | Mean ± SD | 95% CI | $P^\wedge$ |
| **Socio-Demographics & Lifestyle** | | | | | |
| **Age** (years) | 30.8 ± 3.6 | 27.8; 33.7 | 30.2 ± 3.0 | 28.9; 31.4 | .653 |
| **Height** (cm) | 163.7 ± 9.6 | 155.7; 171.7 | 162.4 ± 6.0 | 159.8; 164.9 | .650 |
| **Weight** (kg) | 70.5 ± 19.4 | 54.3; 86.7 | 67.8 ± 13.6 | 62.1; 73.5 | .666 |
| **Body Mass Index** (kg/m$^2$) | 26.1 ± 5.9 | 21.1; 31.0 | 25.9 ± 5.7 | 23.5; 28.3 | .937 |
| **Waist Circumference** (cm) | 82.0 ± 16.7 | 68.0; 96.0 | 82.2 ± 13.5 | 76.5; 87.9 | .968 |
| **Waist to Height Ratio** | 0.5 ± 0.1 | 0.4; 0.6 | 0.5 ± 0.1 | 0.5; 0.5 | .830 |
| **Walking** (minutes/week) | 322.5 ± 403.1 | -14.5; 659.5 | 334.4 ± 260.1 | 224.6; 444.2 | .404* |
| **Reproductive** | | | | | |
| **Age at Menarche** | 13.4 ± 1.9 | 11.8–15.0 | 12.1 ± 1.9 | 11.3–12.9 | .121 |
| **Months of CHC use** | 45.6 ± 37.9 | 10.5–80.6 | 53.2 ± 37.7 | 34.5–72.0 | .653 |
| **Menstrual Cycle Length** (days) | 38.8 ± 29.4 | 14.2; 63.4 | 30.2 ± 3.1 | 28.9; 31.5 | .593* |
| **Luteal Phase Length** (days) | 2.8 ± 4.7 | - 1.5; 7.1 | 6.1 ± 5.2 | 3.9; 8.3 | .167* |

^Independent Sample T-test

*Mann-Whitney test

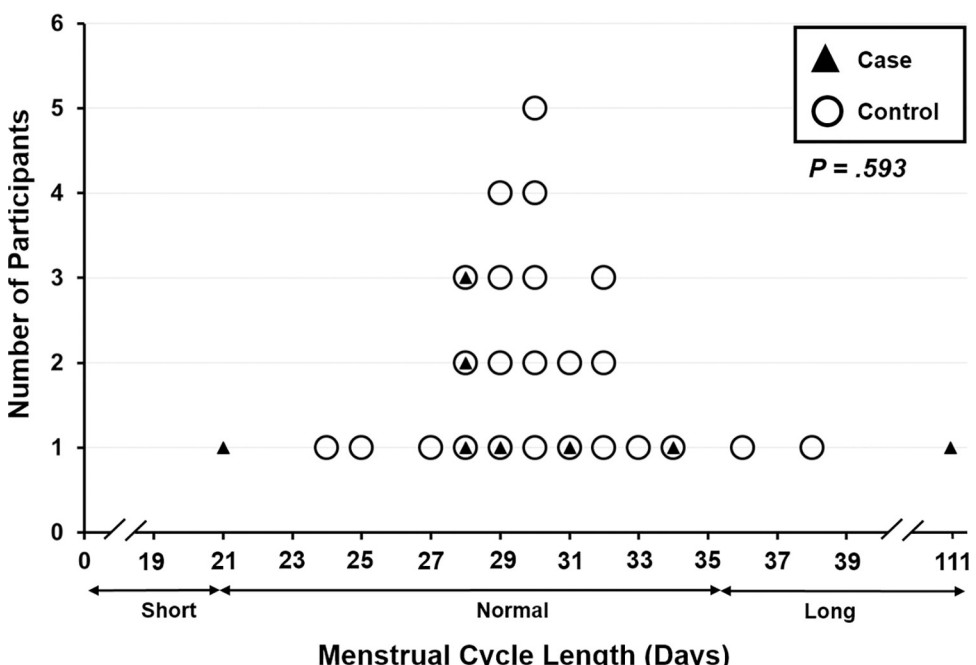

**Fig 1. Lengths of menstrual cycles (in days) for cases having predictably cycling PCOS (n = 8) and controls (n = 24).**

**Table 3,** reporting the entire Diary dataset for both the controls and cases with predictably cycling PCOS, showed there were no menstruation-related differences such as the duration or amount of menstrual flow, menstrual cramps including the Cramps Score and cervical mucus. None of the variables in the top section of the Diary differed.

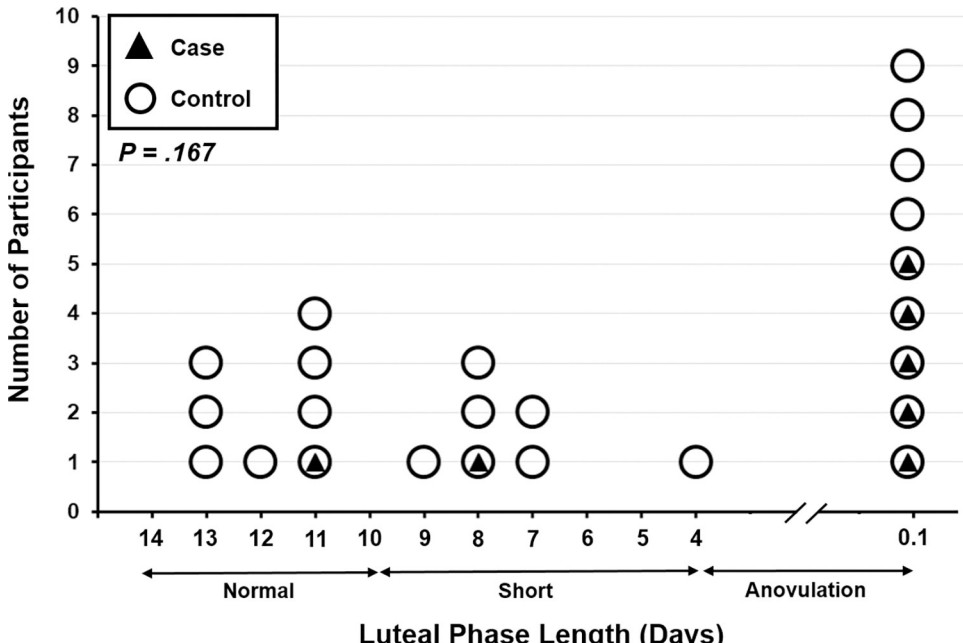

**Fig 2. Lengths of luteal phases (in days) for participants in cases with predictably cycling PCOS (n = 7) and control (n = 24) groups.** Footnote. Ovulation could not be assessed for one of the eight individuals with PCOS.

**Table 3. Menstrual Cycle Diary Information from participants with about month-apart menstrual cycles including Polycystic Ovary Syndrome compared with Age and body mass index-matched controls (1:3) in the Menstruation Ovulation Study 2 during the pandemic.**

| | PCOS (N = 8) | | Control (N = 24) | | Significance |
|---|---|---|---|---|---|
| | **Mean ± SD** | **95% CI** | **Mean ± SD** | **95% CI** | **P†** |
| **Flow Duration** (days) | 6.0 ± 1.1 | 5.1; 6.9 | 5.5 ± 1.4 | 4.9; 6.1 | .454 |
| **Flow Amount** (mean flow/days of flow in ml/day) | 11.4 ± 4.2 | 7.9; 14.3 | 14.1 ± 6.1 | 11.5; 16.6 | .237 |
| **Cramp Duration** (days) | 7.6 ± 7.2 | 1.6; 13.6 | 4.9 ± 4.7 | 2.9; 6.9 | .535 |
| **Cramp Intensity** (0–4) (intensity/days with cramps) | 1.7 ± 0.7 | 1.1; 2.2 | 1.7 ± 0.7 | 1.3; 2.0 | .881 |
| **Cramp Score** (duration x mean intensity) | 13.9 ± 17.2 | - 0.5; 28.3 | 8.5 ± 7.7 | 5.2; 11.7 | .593 |
| **Front-Breast Tenderness** (0–4) | 1.2 ± 0.3 | 0.8; 1.6 | 1.4 ± 0.6 | 1.1; 1.7 | .491 |
| **Side-Breast Tenderness** (0–4) | 1.5 ± 0.5 | 0.9; 2.1 | 1.6 ± 0.6 | 1.3; 2.0 | .933 |
| **Fluid Retention** (0–4) | 1.4 ± 0.4 | 1.1; 1.8 | 1.5 ± 0.6 | 1.2; 1.8 | .917 |
| **Mucus Secretions** (0–4) | 1.7 ± 0.8 | 1.1; 2.4 | 1.4 ± 0.3 | 1.3; 1.5 | .339 |
| **Constipation** (0–4) | 1.4 ± 0.5 | 1.0; 1.9 | 1.2 ± 0.2 | 1.0; 1.3 | .208 |
| **Headache** (0–4) | 1.6 ± 0.5 | 1.3; 2.0 | 1.5 ± 0.5 | 1.3; 1.8 | .566 |
| **Sleep Problems** (0–4) | 1.9 ± 0.5 | 1.5; 2.3 | 1.6 ± 0.5 | 1.4; 1.8 | .145 |
| **Feeling of Frustration** (0–4) | 1.7 ± 0.6 | 1.1; 2.3 | 1.6 ± 0.5 | 1.4; 1.8 | .924 |
| **Feeling of Depression** (0–4) | 1.6 ± 0.3 | 1.2; 2.0 | 1.6 ± 0.6 | 1.2; 1.9 | .649 |
| **Feeling of Anxiety** (0–4) | 1.9 ± 0.9 | 1.1; 2.7 | 1.5 ± 0.4 | 1.3; 1.7 | .174 |
| **Appetite** (1–5) | 3.0 ± 0.3 | 2.8; 3.2 | 3.1 ± 0.3 | 2.9; 3.2 | .654 |
| **Breast Size** (1–5) | 3.0 ± 0.2 | 2.8; 3.2 | 3.2 ± 0.3 | 3.0; 3.3 | .245 |
| **Interest in Sex** (1–5) | 3.0 ± 0.2 | 2.8; 3.1 | 3.2 ± 0.4 | 3.0; 3.3 | .118 |
| **Feeling of Energy** (1–5) | 2.7 ± 0.3 | 2.5; 2.9 | 2.9 ± 0.4 | 2.7; 3.0 | .611 |
| **Feeling of Self-Worth** (1–5) | 2.9 ± 0.2 | 2.7; 3.1 | 3.0 ± 0.4 | 2.9; 3.2 | .471 |
| **Outside Stresses** (1–5) | 3.4 ± 0.2 | 3.2; 3.6 | 3.0 ± 0.5 | 2.8; 3.3 | **.020** |

†Mann-Whitney test

In the lower portion of the Diary "usual" recorded experiences related to appetite, breast size, interest in sex and feeling of energy were also not statistically different. However, Outside Stresses were significantly higher in women diagnosed with PCOS compared with controls ($P$ = .020) despite the milder and predictably cycling characteristics of those with PCOS.

## Discussion

To our knowledge, these are the first case-control data in individuals living with polycystic ovary syndrome (PCOS) that examined their everyday and menstrual cycle related experiences compared with controls closely matched for age and BMI. Note that all participants in this cohort study were only eligible if they had predictable, approximately month-apart cycles, thus, the women with PCOS likely were less symptomatic than the majority with this diagnosis. The cohorts were similar in gynecological ages, exercise, alcohol use, education and occupation. Thus, not only the matching but the fact that all were from a community-dwelling cohort (without any patients or medical-help-seeking women) means that they are very similar groups. This comparison was unusual because the women with PCOS and the controls had similar cycle lengths (due to enrolment criteria) and ovulatory characteristics (likely because of the effects of the SARS-CoV-2 pandemic on ovulatory characteristics in the MOS2 control participants [36]). This unique information is the beginning of collection of data that may eventually benefit women who are living with PCOS, their healthcare providers, PCOS resource creators and advocacy groups.

Some individuals with PCOS would have self-excluded because of too-long cycles or amenorrhea. In addition, cycle lengths were likely to be similar because the women with PCOS in this study were in their early 30s. It is known that increased ages within the premenopausal years are associated with improved normality of reproduction in women with PCOS [37]. Thus, cycle lengths are likely similar because of the increased regularity of cycles in women with PCOS as they become older pre/perimenopausal women [3].

Surprisingly, women living with PCOS also had similar luteal phase lengths as controls. It would normally be expected that ovulatory disturbances would be more common in women with PCOS than in community dwelling controls. However, in the **overall MOS2 data**, we found that 63% of all participants (n = 108) experienced ovulatory disturbances [36]. We attribute these highly prevalent ovulatory disturbances in the entire cohort to the multidimensional stressors of living through SARS-CoV-2 [36]. The surprisingly similar ovulatory characteristics in PCOS may also be part of the phenotypic PCOS reproductive changes reported in the 30s and 40s. A 21-year prospective study documented that ovulatory cycles increased within-woman with PCOS over that period [38]. Ovulatory disturbances were unlikely related to acute COVID-19-related illnesses, nor to SARS-CoV-2 vaccinations since neither occurred during the study cycle for any participants.

Our pilot study was limited to a cohort of what turned out to be highly educated women in their early 30s living within Metro Vancouver, B.C who were willing and able to volunteer during the early days of the SARS-CoV-2 pandemic. One possible explanation for the similar levels of anxiety and depression in women with PCOS and matched controls we observed may be that women with PCOS had predictable cycles and likely lesser impacts of this condition on their lives. Alternatively, higher education may have been a protective factor although it did not protect from the adaptive changes in ovulation we documented in this whole cohort. However, our study highlights the need for rigorous and high-quality quantitative and qualitative methods to better understand the experiences of women with a whole spectrum of experiences with PCOS.

This pilot study has some limitations. One limitation is the small sample size dictated by the proportion of women with physician-diagnosed PCOS enrolled in the MOS2 study. It is less than the expected 10% of PORA with PCOS because of our enrolment criteria requiring both regular cycles and not using hormonal contraceptives. Combined hormonal contraceptives are the current standard of care for PCOS [11]. Thus, the inclusion and exclusion criteria for MOS2 limited the eligible PCOS participants. This analysis was a pre-planned secondary objective of the overall MOS2 protocol.

Another, historically accepted limitation of our study is that, by questionnaire women self-reported PCOS, which could lead to inaccuracy. We doubt that is true based on a prospective community-based cohort study conducted from 2006–2010, in which Chan *et al.* [39] tested a screening telephone questionnaire for PCOS. This consisted of three questions to assess the accuracy of self-reported PCOS, androgen excess (AE), and irregular menses in women aged 14 to 45 years [39]. Participants were asked to self-assess the presence/absence of male-like hair patterns and menstrual irregularity and were then invited to undergo a direct examination, including completing a medical history and modified Ferriman-Gallwey hirsutism score, ovarian ultrasound, and measurement of circulating total and free testosterone, DHEAS, TSH, prolactin, and 17-hydroxyprogesterone levels [39]. Participants with self-assessed irregular menses and/or excess hair were labeled "Possible Androgen Excess (Poss-AE)" and those self-assessed with regular menses and no excess hair were labeled "Probable Non-Androgen Excess (Non-AE)." The study was completed in 206/298 (69%) of the Poss-AE and in 139/192 (73%) of the Non-AE. Of Poss-AE and Non-AE subjects, 82.5% and 15.8%, respectively, presented with PCOS. The calculated sensitivity, specificity, positive and negative predictive values of the

3-question telephone survey to predict PCOS were 89%, 78%, 85%, and 83%, respectively [39]. Overall, that study suggested women's self-report of PCOS is reliable. That we also specified a physician-diagnosis of PCOS likely increased that accuracy.

Despite these limitations, this pilot has many strengths. We were able to successfully match cases with controls for both age and BMI thus our first feasibility objective was met. We have also successfully described the menstrual cycle and daily experiences of women with PCOS and predictable menstrual cycles compared with age- and BMI- similar controls.

All the participants were from a community-dwelling cohort and were similarly recruited. All shared the experiences of the pandemic lockdown, lack of access to extended family and close friends, health insecurity, and few medical screening facilities. Later in recruitment, all women also had to make similar decisions about vaccines. Although small sample sizes may limit the generalizability of findings, descriptive data from small samples can provide valuable initial insights into a research question, generate hypotheses, and inform the design of future studies—all are purposes of a case-control design. Specifically, women living with PCOS were not only carefully matched for age and BMI, but they were also similar to controls in terms of all our extensively documented sociodemographic, educational, lifestyle, and other relevant characteristics (**Table 1**). This investigation was a planned secondary objective in a pre-published, ethics approved online protocol for the MOS2 study and thus not biased by something we observed with the MOS2 cohort during or following data collection. Finally, and most importantly we used standard, scientific case-control methodology and successfully matched all cases in a 1:3 ratio with controls. In a condition such as PCOS in which few experiential data have been reported, gathering even a small amount of descriptive data can help to provide a deeper understanding of the daily lives of those living with the condition. This information can be used to guide the development of more targeted and effective interventions and resources for patients, and to direct future research that can further explore and confirm the initial findings in larger cohorts.

In addition, this MOS2 and current case-control study employed reliable and validated tools to document women's demographics, anthropomorphic measures, menstrual cycle lengths, and ovulatory characteristics. These included the CaMos© questionnaire, Menstrual Cycle Diary©, and Quantitative Basal Temperature© ovulation and luteal length statistical analysis method, which were the same for both groups, thus minimizing any potential bias. Additionally, the use of daily Diary reports, rather than questionnaire data, is a significant strength as it mitigates the risk of recall bias. Note that often diary data are used to validate questionnaire-collected information. Overall, the rigorously designed study methods and the use of reliable tools and techniques in this pilot study provide evidence that a more definitive study would be feasible.

## Conclusions

This first pilot case-control study of the menstrual cycle and daily life experiences of women with PCOS who had predictable, month-apart cycles compared with age- and BMI-similar community-based controls showed that studying the experiences of women with PCOS in this way is feasible. Women with PCOS were not more likely to experience ovulatory disturbances than controls, likely because these data were all collected within a time of major social/economic/health challenge during which controls also experienced a high prevalence of ovulatory disturbances. The results of this pilot study support the feasibility of a necessary and much larger case-control or experimental study of the daily and menstrual cycle experiences of women living with PCOS.

## Acknowledgments

Health Canada funded the MOS2 contract to study the menstrual cycles of community-dwelling women and to collect urines for environmental contaminants.

## Author Contributions

**Conceptualization:** Kaitlin Nelson, Sonia Shirin, Jerilynn C. Prior.

**Formal analysis:** Sonia Shirin.

**Investigation:** Sonia Shirin, Dharani Kalidasan, Jerilynn C. Prior.

**Methodology:** Kaitlin Nelson, Sonia Shirin, Jerilynn C. Prior.

**Project administration:** Dharani Kalidasan.

**Software:** Sonia Shirin.

**Supervision:** Jerilynn C. Prior.

**Visualization:** Dharani Kalidasan.

**Writing – original draft:** Kaitlin Nelson.

**Writing – review & editing:** Kaitlin Nelson, Sonia Shirin, Dharani Kalidasan, Jerilynn C. Prior.

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
