## [Decision Letter · Decision Letter 0]

26 Sep 2023

PONE-D-23-20996Experiences of women living with Polycystic Ovary Syndrome: A pilot case-control, single-cycle, daily Menstrual Cycle Diary© study during the SARS-CoV-2 pandemicPLOS ONE

Dear Dr. Prior,

Thank you for submitting your manuscript to PLOS ONE. After careful consideration, we feel that it has merit but does not fully meet PLOS ONE’s publication criteria as it currently stands. Therefore, we invite you to submit a revised version of the manuscript that addresses the points raised during the review process.

The reviewers have some concerns about the group comparisons; please, see the reviewers comments below.

We look forward to receiving your revised manuscript.

Kind regards,

Fabio Vasconcellos Comim, MD,PhD

Academic Editor

PLOS ONE

Journal Requirements:

Reviewers' comments:

Reviewer's Responses to Questions

**Comments to the Author**

1. Is the manuscript technically sound, and do the data support the conclusions?

Reviewer #1: No

Reviewer #2: Partly

2. Has the statistical analysis been performed appropriately and rigorously? 

Reviewer #1: Yes

Reviewer #2: No

3. Have the authors made all data underlying the findings in their manuscript fully available?

Reviewer #1: Yes

Reviewer #2: Yes

4. Is the manuscript presented in an intelligible fashion and written in standard English?

Reviewer #1: No

Reviewer #2: Yes

5. Review Comments to the Author

Reviewer #1: This research article sought to examine the menstrual cycle and daily life experiences of women with PCOS as compared to women without PCOS during the Covid-19 pandemic. PCOS is an important and needed area of research. However, this article presents several flaws.

First, requiring month-apart menstrual cycles severely limited your PCOS population, especially those with most disturbing PCOS symptoms. This study most likely sampled the milder phenotype (a much smaller prevalence), and oligo- ovulatory women with PCOS exhibit milder symptoms. There is also research showing that depressive levels among this phenotype are comparable to women without PCOS due to lack of or less obesity, hirsutism, alopecia, etc. This was not a true comparison of women with PCOS in its most prevalent form and women without PCOS.

As such, the conclusions are misrepresented and overstated. Most women with PCOS (phenotypes A and B) do not have similar cycle lengths and ovulatory characteristics. This needs to be stated in the intro and conclusion given the chosen group of women with PCOS. Anovulatory women with PCOS are more often amenorrheic and present with longer cycle duration.

The authors state this is a pilot study, but there are no measures of feasibility and acceptability. Although this study included a control group, it is more of a cross-sectional design due to its results of prevalence, not predictors of outcome measures.

Self-worth, as a concept, was not properly defined. As written, the study measured self-image. Outside stresses as defined by the MOS2 was not specified. There is only a signal of significant difference on outside stressors. Without this definition, the discussion stretches beyond the results. Also, there is a notable difference between statistical and clinical significance. This study has not indication or description of clinical significance, especially among those who represent about 4% of women with PCOS. Thus, this study severely lacks external validity.

Lines 363: Information introduced in the discussion that should be placed in the background information. Also, the purpose of the study was not a comparison of stressful situations (i.e., number or type) and stressful situations were not defined (as stated above). Also, it is discussed that coping mechanisms mitigate the relationship between stress and mental disorders. Coping mechanisms were not measured. Types of stress were not measured. There was no difference detected with symptoms of anxiety and depression.

Lines 469-475 of the conclusion are irrelevant to the study. The women with PCOS were similar because women with PCOS similar to those without PCOS were purposefully studied. Feasibility was not supported because it was not measured.

Reviewer #2: Title: Experiences of women living with Polycystic Ovary Syndrome: A pilot case-control, single-cycle, daily Menstrual Cycle Diary© study during the SARS-CoV-2 pandemic

In this single-cycle prospective case-control study authors have examined the menstrual cycle and daily life experiences in PCOS patients compared with cohort controls using data from the Menstruation & Ovulation Study 2 (MOS2) during the first 1.5 years of SARS-CoV-2 pandemic. Authors have applied adequate methods in their analysis and results are interesting. Although the manuscript is quiet interesting, there are several serious concerns that should be addressed. Here are my comments:

1. Sample power: Was the sample power calculated? Despite the number of subjects studied the sample power needs to be calculated.

2. Statistical analysis: Authors should perform fit Chi-square calculation to determine ‘P’ value and Pooled odds ratios (OR) and 95% confidence intervals (CIs) for all the parameters they analyzed.

3. All the statistical data must be tested for normality and group variance prior to subject statistical analysis.

4. Significant values should be corrected for the number of comparisons performed by suitable post hoc tests (example: Bonferroni method)

5. Ethnicity of study population: Authors should also present their data Ethnicity wise to strengthen their observations.

6. PLOS authors have the option to publish the peer review history of their article (what does this mean?). If published, this will include your full peer review and any attached files.

Reviewer #1: **Yes: **Pamela J. Wright

Reviewer #2: **Yes: **Suresh Govatati

---

## [Author Response · Author response to Decision Letter 0]

17 Nov 2023

November 1, 2023

Fabio Vasconcellos Comim MD, PhD

PLOS ONE Academic Editor

Dear Dr. Fabio Comim:

Re: Revision of “Experiences of women living with Polycystic Ovary Syndrome: A pilot case-control, single-cycle, daily Menstrual Cycle Diary study during the SARS-CoV-2 pandemic

Thank you for providing two the reviews. 

We will first respond to your editorial issues before responding individually to each of the reviewer’s questions and comments.

1. To the best of our knowledge, this manuscript meets PLOS ONE style requirements 

2. Availability of data—we are unable to publish the original data from this case-control study for two reasons: 

--first because it is nested within a cohort study. That whole cohort study has not yet been published; 

--second because we do not have ethical approval for release of individual participant data. The consent specifies publication of aggregate data in a peer reviewed report. 

Therefore, we will make data available to interested and qualified investigators who request it from the principal investigator. 

3. We have removed the sentence preceding “data not shown” each of the three times that we wrote that. 

4. We have removed the reference to ethical approval that followed the body of the manuscript.

Responses to reviewers

 We will quote the reviewer’s comment.

 Our response will be in italics

 When we have changed the manuscript, we will use this format: Line# (in the track change, rather than the original submitted document) revised to “. . . . .” 

Reviewer 1:

Thank you for acknowledging that PCOS is an important area of study and stating we’ve correctly performed the statistical analyses. 

We have used track changes to correct conflicts in tense and other grammatical errors that previously were present. We trust we’ve now produced an RI that is well-written.

This research article sought to examine the menstrual cycle and daily life experiences of women with PCOS as compared to women without PCOS during the Covid-19 pandemic. PCOS is an important and needed area of research. 

Thank you for your affirmation of the importance of PCOS research. We agree with your statement in a recent paper that it is important for research to be “multidisciplinary, multidimensional, and multi-level to ameliorate biopsychosocial issues”. Ours is an initial attempt to do exactly that.

However, this article presents several flaws. First, requiring month-apart menstrual cycles severely limited your PCOS population, especially those with most disturbing PCOS symptoms. 

You are quite right that predictable month-apart menstrual cycles likely implied that these women would have less intense PCOS symptoms. We have now acknowledged that in multiple places in the manuscript as noted below 

Abstract Line # 39, added “having approximately month-apart cycles”

Abstract Lines #59-60, added “in women with milder PCOS”

Lines #144-145, Introduction about PCOS “even though it may be less symptomatic than in those with oligo-amenorrhea”

Line #206, Methods add “normally cycling” PCOS

Line #274 in Results, added, “normally cycling and milder” PCOS

Lines 302-3, title of Table 1, added “all of whom had approximately month-apart cycles” 

Lines 309-10 in describing Figure 1 that shows similar cycle lengths, “(likely because needed to have approximately month-apart cycles to be eligible).” 

Lines 318-added “about predictably cycling” PCOS

Lines 323, about ovulation, added “in women with normally cycling PCOS, luteal phase lengths. . . 

Lines 332, added “predictably cycling” PCOS

Lines 336-337, added “about month-apart menstrual cycles including” PCOS and controls. . .

Lines 347, end of Results, added “despite the milder and predictably cycling characteristics of those with PCOS”

Lines 350-53, second sentence of Discussion, added “Note that all participants in this cohort study were only eligible if they had predictable, approximately month-apart cycles, thus the women with PCOS likely were less symptomatic than the majority with this diagnosis.” 

Line 360, cycle lengths “(due to enrolment criteria)”

Lines 425-6, explaining similar depression “that women with PCOS had predictable cycles and likely lesser impacts of this condition on their lives.”

Line 502 Conclusion women with PCOS “who had predictable, month-apart cycles”

This study most likely sampled the milder phenotype (a much smaller prevalence), and oligo- ovulatory women with PCOS exhibit milder symptoms. There is also research showing that depressive levels among this phenotype are comparable to women without PCOS due to lack of or less obesity, hirsutism, alopecia, etc. This was not a true comparison of women with PCOS in its most prevalent form and women without PCOS.

Thank you.

We agree that the larger cohort study in which this case-control investigation nested, with its requirement for about monthly periods, would have eliminated women with PCOS having amenorrhea or oligomenorrhea. 

However, the natural history of PCOS across women’s life cycles is for oligomenorrhea to decrease with increasing age [1] and cycles to become shorter and more regular and even to develop more normal ovulation[2]. It is of interest that the women with PCOS in our case-control study, although they could have been between 19-35 years of age, were all 26 years of age or older. Thus, although they represent a milder phenotype, they are still an accurate reflection of the experiences of some women with PCOS.

As such, the conclusions are misrepresented and overstated. Most women with PCOS (phenotypes A and B) do not have similar cycle lengths and ovulatory characteristics. This needs to be stated in the intro and conclusion given the chosen group of women with PCOS. Anovulatory women with PCOS are more often amenorrheic and present with longer cycle duration.

We have altered the introduction as requested by adding this sentence: 

Lines 87-89-“However, we know little of the daily experiences of individuals with PCOS, especially those whose cycles have become more predictable and month-apart.”

We have also modified the conclusions as shown on line 502 (see above) to specify that they apply only to women with PCOS who initially had predictable, month-apart cycles.

The authors state this is a pilot study, but there are no measures of feasibility and acceptability. 

You are quite right that, as a pilot study, it should have outcomes that relate to feasibility. We had these questions about feasibility but had previously not stated them. 

Lines 158-163, added Our objectives in this pilot study “related to feasibility: 1) To determine, in a case-control study, if each woman with PCOS could effectively be matched with three age- and BMI-matched control women within a larger community-dwelling cohort; 2) To describe the menstrual cycle and daily experiences of women with predictably cycling PCOS compared with matched controls all of whom recorded daily in the standardized Menstrual Cycle Diary© [19].”

Also, given that pilot studies do not normally include hypotheses (which are generally reserved for experimental designs) we have deleted the hypotheses from this manuscript[3] . 

Although this study included a control group, it is more of a cross-sectional design due to its results of prevalence, not predictors of outcome measures.

I agree that the original presentation of our design was misleading. 

We disagree. We believe it qualifies as a case-control type of observational design because all cases were selected to have PCOS. It also differs from a cross-sectional design in that the controls without the condition of interest were matched on key variables with members of the rest of the cohort. 

Self-worth, as a concept, was not properly defined.

Self-worth is defined as an individual’s personal evaluation of themselves as of value. But there is debate in the literature about whether self-worth is a synonym of self-esteem, or further is a psychological state or a trait. You are right that we did not define it. It is beyond the scope of this pilot study to describe how we advised women to record each of the items in the Menstrual Cycle Diary©. Instead we have provided a link to a video that instructs participants in trials.

Outside stresses as defined by the MOS2 was not specified. There is only a signal of significant difference on outside stressors. Without this definition, the discussion stretches beyond the results. 

We respond similarly as to self-worth above—defining all of these Diary items is not necessary. In particular, we have removed differences in perceived outside stress from the discussion. 

Also, there is a notable difference between statistical and clinical significance.

We agree—we have removed that terminology from the note about how we designated probability in the statistical evaluation section. 

This study has not indication or description of clinical significance, especially among those who represent about 4% of women with PCOS. Thus, this study severely lacks external validity.

As mentioned, we’ve removed any discussion of clinical significance from this document—we are only making descriptive observations. 

The 8 women with physician-documented PCOS and month-apart cycles are 7.4% of the 108 women with complete data in the MOS2 cohort. We have no way of knowing what proportion of all women in the population diagnosed with PCOS those with predictable, approximately month-apart cycles may be, however. 

Lines 363: Information introduced in the discussion that should be placed in the background information. 

Thank you. We have moved these paragraphs to the introduction.

Also, the purpose of the study was not a comparison of stressful situations (i.e., number or type) and stressful situations were not defined (as stated above). Also, it is discussed that coping mechanisms mitigate the relationship between stress and mental disorders. Coping mechanisms were not measured. Types of stress were not measured. There was no difference detected with symptoms of anxiety and depression.

We agree and no longer have a quantitative comparison of the results in the two groups as part of our aims, hypotheses, discussion nor our conclusions. 

Lines 469-475 of the conclusion are irrelevant to the study. The women with PCOS were similar because women with PCOS similar to those without PCOS were purposefully studied. 

We agree. We have removed discussion of quantitative differences between groups. 

Lines 532-535 “This first pilot case-control study of the menstrual cycle and daily life experiences of women with PCOS who had predictable, month-apart cycles compared with age- and BMI-similar community-based controls showed that studying the experiences of women with PCOS in this way is feasible.” 

Feasibility was not supported because it was not measured.

As mentioned earlier, we had feasibility objectives that now are clearly stated. 

Thank you for your through and thoughtful review of this manuscript.

Reviewer 2:

Thank you for your assessment of this manuscript, acknowledging that the topic is of interest and also that it is well-written. 

1. Sample power: Was the sample power calculated? Despite the number of subjects studied the sample power needs to be calculated.

Thank you for your question. Sample size is obviously necessary for experimental studies. 

We apologize that our manuscript was originally written in a confusing way that suggested an experimental nature. However, it is a pilot study.[3] It is our understanding that pilot studies for RCTs may include a sample power calculation if one is possible. However, this was a case-control observational study which does not require this. Plus, since this is the first such investigation, we had no way of even estimating sample size.

2. Statistical analysis: Authors should perform fit Chi-square calculation to determine ‘P’ value and Pooled odds ratios (OR) and 95% confidence intervals (CIs) for all the parameters they analyzed.

Thank you. We did perform Chi-square calculations but since this is a pilot study [3] that we have now revised to more appropriately include feasibility rather than experimental objectives, further analyses such as odd ratios are neither appropriate, nor likely to be informative with such small numbers in the PCOS group.

3. All the statistical data must be tested for normality and group variance prior to subject statistical analysis.

Unfortunately, the order of paragraphs within the statistical evaluation section was not appropriate. We’ve now moved that statement to the be the first. Which, is what we did—as you so rightly say, test for normality of distribution and group variance. 

4. Significant values should be corrected for the number of comparisons performed by suitable post hoc tests (example: Bonferroni method)

You are quite right that, with the many items in the Menstrual Cycle Diary plus variables we’ve collected to assess the socio-cultural, educations and ethnic distributions of both groups, in an experimental study, some correction for many tests would be needed. However, this was a pilot, observational study that, as revised only has appropriate feasibility rather than any experimental objectives. 

5. Ethnicity of study population: Authors should also present their data Ethnicity wise to strengthen their observations.

We’re sorry, but it is not quite clear what you are asking us to do. 

We found no differences in ethnicity between the cases with physician-diagnosed PCOS plus predictable, month-apart cycles and age- and BMI-matched controls (Table 1). 

It is not clear what would be gained by decreasing our already small participant numbers further by ethnicity-specific analyses. 

Thank you for your careful review of this manuscript which for which we have made major revisions so it is appropriately presented as a pilot case-control observational rather than an experimental study. 

Sincerely,

Jerilynn C. Prior on behalf of all authors

References cited in responses to this review

1. Elting, M.W., et al., Women with polycystic ovary syndrome gain regular menstrual cycles when ageing. Human Reproduction, 2000. 15(1): p. 24-28.

2. Schmidt, J., et al., Reproductive hormone levels and anthropometry in postmenopausal women with polycystic ovary syndrome (PCOS): a 21-year follow-up study of women diagnosed with PCOS around 50 years ago and their age-matched controls. J Clin Endocrinol.Metab, 2011. 96(7): p. 2178-2185.

3. Thabane, L., et al., A tutorial on pilot studies: the what, why and how. Bmc Medical Research Methodology, 2010. 10.

---

## [Decision Letter · Decision Letter 1]

11 Dec 2023

Experiences of women living with Polycystic Ovary Syndrome: A pilot case-control, single-cycle, daily Menstrual Cycle Diary© study during the SARS-CoV-2 pandemic

PONE-D-23-20996R1

Dear Dr. Prior,

We’re pleased to inform you that your manuscript has been judged scientifically suitable for publication and will be formally accepted for publication once it meets all outstanding technical requirements.

Kind regards,

Fabio Vasconcellos Comim, MD,PhD

Academic Editor

PLOS ONE

**Comments to the Author**

1. If the authors have adequately addressed your comments raised in a previous round of review and you feel that this manuscript is now acceptable for publication, you may indicate that here to bypass the “Comments to the Author” section, enter your conflict of interest statement in the “Confidential to Editor” section, and submit your "Accept" recommendation.

Reviewer #3: All comments have been addressed

2. Is the manuscript technically sound, and do the data support the conclusions?

Reviewer #3: Yes

3. Has the statistical analysis been performed appropriately and rigorously? 

Reviewer #3: Yes

4. Have the authors made all data underlying the findings in their manuscript fully available?

Reviewer #3: Yes

5. Is the manuscript presented in an intelligible fashion and written in standard English?

Reviewer #3: Yes

6. Review Comments to the Author

Reviewer #3: The present study is a case-control study that compares the menstrual cycle and daily life experiences in women with PCOS with age and BMI-matched controls during the first 1.5 years of the SARS-CoV-2 pandemic. The authors used information from the Menstruation & Ovulation Study 2 (MOS2). The study shows no differences in ovulatory disturbances between the two groups. Although the study has a small sample size, it adds to the current literature the feasibility of studying the subject.

I have carefully revised the answers to the previous reviewers' queries and am happy about the responses.

7. PLOS authors have the option to publish the peer review history of their article (what does this mean?). If published, this will include your full peer review and any attached files.

Reviewer #3: No

---

## [Editor Report · Acceptance letter]

19 Dec 2023

PONE-D-23-20996R1 

PLOS ONE

Dear Dr. Prior, 

I'm pleased to inform you that your manuscript has been deemed suitable for publication in PLOS ONE. Congratulations! Your manuscript is now being handed over to our production team.

Kind regards, 

on behalf of

Prof Fabio Vasconcellos Comim 

Academic Editor

PLOS ONE